# Addressing the Challenges and Barriers to the Integration of Machine Learning into Clinical Practice: An Innovative Method to Hybrid Human–Machine Intelligence

**DOI:** 10.3390/s22218313

**Published:** 2022-10-29

**Authors:** Chadia Ed-Driouch, Franck Mars, Pierre-Antoine Gourraud, Cédric Dumas

**Affiliations:** 1École Centrale Nantes, IMT Atlantique, Nantes Université, CNRS, LS2N, UMR 6004, F-44000 Nantes, France; 2Centrale Nantes, Nantes Université, CNRS, LS2N, UMR 6004, F-44000 Nantes, France; 3Clinique des Données, Pôle Hospitalo-Universitaire 11: Santé Publique, CHU Nantes, Nantes Université, INSERM, CIC 1413, F-44000 Nantes, France; 4Département Automatique, Productique et Informatique, IMT Atlantique, CNRS, LS2N, UMR CNRS 6004, F-44000 Nantes, France

**Keywords:** human–machine collaboration, machine learning, physician–algorithm collaboration, clinical decision-making, personalized medicine, multiple sclerosis

## Abstract

Machine learning (ML) models have proven their potential in acquiring and analyzing large amounts of data to help solve real-world, complex problems. Their use in healthcare is expected to help physicians make diagnoses, prognoses, treatment decisions, and disease outcome predictions. However, ML solutions are not currently deployed in most healthcare systems. One of the main reasons for this is the provenance, transparency, and clinical utility of the training data. Physicians reject ML solutions if they are not at least based on accurate data and do not clearly include the decision-making process used in clinical practice. In this paper, we present a hybrid human–machine intelligence method to create predictive models driven by clinical practice. We promote the use of quality-approved data and the inclusion of physician reasoning in the ML process. Instead of training the ML algorithms on the given data to create predictive models (conventional method), we propose to pre-categorize the data according to the expert physicians’ knowledge and experience. Comparing the results of the conventional method of ML learning versus the hybrid physician–algorithm method showed that the models based on the latter can perform better. Physicians’ engagement is the most promising condition for the safe and innovative use of ML in healthcare.

## 1. Introduction

The complexity of chronic diseases has required a shift from a disease-centered focus to a more patient-centered approach. This approach remains a challenge to identify the patients’ physiological dysfunctions, individual disease form, and underlying causes relative to their situation, features, and desired outcomes. A patient-centered approach represents the most appropriate strategy to provide a more personalized decision protocol for each patient.

Given the large amounts of patient data available (cohorts, clinical trials, etc.), the developed data storage infrastructures, and the advances in data science, artificial intelligence (AI) has become the most targeted tool that offers an enabling environment to address the challenge of personalized medical decisions [1]. Machine learning (ML) is one of the core components of AI applications in healthcare. The flexibility and proficiency of machine learning algorithms with available computing power have a special role to play in advancing personalized medicine. They are able to detect patterns, differences, and correlations within the data to make predictions about the likelihood of uncertain outcomes in complex care settings. ML models have been used as a supporting tool to identify specific diseases, disorders [2], and tumor types from the analysis of clinical/pathological data [3]. ML-based approaches have been demonstrated useful in developing polygenic risk scores that can help to identify individuals at high genetic risk for a disease [4]. ML has also proven useful in figuring out how medications and compounds can impact a cell’s features [5]. However, the adoption of ML approaches in healthcare is facing complex challenges.

Researchers have identified two main issues for the adoption of ML in healthcare systems [6]: technical concerns such as data quality and the ‘black box’ problem, and human concerns regarding the ethics, legality, and acceptability of ML.

The quality of the training data is correlated to the quality of the obtained results. If this data is incomplete or reflects only a part of the population, the resulting model will be relevant only for the population represented in the dataset, which may have unintended adverse effects on the patient and clinician. Transparency, simplification, understanding, and clinical utility of the datasets must be emphasized, and each result must be questioned for its clinical applicability.

ML algorithms can provide reliable learning models if the quality of the used data is suitably sufficient. However, these models are difficult to understand for a physician, who needs to know how a model translates input data into output decisions. This ‘black box’ problem can be problematic and lead to a lack of confidence or a sense of concern and controversy about the reliability of ML. These gaps in knowledge of the model itself and understanding of the learning process [7] require that the results of the algorithms be clearly included in the decision-making process of medical practice. Furthermore, more importance should be given to the explanation of the results obtained by the ML models. It is also a legal requirement [8].

The inability of predictions to be transparent, clear, or trusted by physicians negatively affects their use. The potential risks related to data and their effect on patient safety and the concern that physicians could be replaced by ML systems have created barriers to the integration of machine learning into clinical practice. As demonstrated by Xiang et al. [9], patients and the public in general trust the medical decisions made by the physician more than by ML systems. Thus, physician engagement is the most promising condition for the safe and innovative use of ML in healthcare. However, many physicians reject ML solutions because they are unsure of the performance and benefits of these solutions [10]. To benefit their patients from AI advances, physicians need to have direct control and solid knowledge of the recommendations of an ML system. They should be able to interact easily with decision support tools and to interpret the findings generated by ML algorithms [11]. To do so, cognitive systems engineering [12] has the potential to support physician–algorithm communication through the design of new ML tools centered on human–machine collaboration [9,13].

The hybridization of human and artificial intelligence has emerged as a new form of human–machine collaboration to enhance each other and bring out the greatest potential of both [14]. This hybrid intelligence (HI) concept consists in merging the mutual intelligence of human and artificial agents in order to overcome the limitations of artificial intelligence [15]. Thus, AI can provide support for human decision-making, or humans can assist the ML process to support AI tasks. In a theoretical approach guided by empirical evidence, L. Hong et al. [16] have shown that humans and algorithms can cooperatively make better predictions than either alone. N. Zheng et al. [17] have also shown that application of cognitive hybrid-intelligence with human–machine interaction can be revolutionary in the medical field. In their recent review of behavioral models of hybrid intelligence, E. Akmeikina et al. [18] pointed out several important issues that are not yet well addressed in the literature. One of the most important issues was about how AI should be designed to provide fair and understandable support for humans.

To address this issue, we propose a new method that allows the integration of empowered humans (expert physicians) in the learning process of ML algorithms. Thus, the prediction model created includes human reasoning. The proposed method promotes the use of quality-approved data to be transformed according to the knowledge and experience of expert physicians and then used to train ML algorithms. To validate our approach, we compare the conventional method of creating prediction models with our HI method.

This method of creating ML models based on medical practice through the integration of the solid knowledge of expert physicians is expected to be the gateway for new applications to support the personalization of clinical decision-making.

## 2. Materials and Methods

### 2.1. Case Study and Data Source

Multiple sclerosis (MS) [19] is a chronic autoimmune inflammatory disease of the central nervous system that affects 2.8 million people worldwide [20]. MS is characterized by variability in terms of symptoms, forms, and arguments for diagnosis, prognosis, and therapeutic follow-up. Through multiple interactions with MS physicians (interviews, observations) and the literature [21,22,23,24], key predictors of MS activity were identified as follows:The worsening of the Expanded Disability Status Scale (EDSS), which is a score to measure the level of physical disability;The presence of new lesions on the MRI (Magnetic Resonance Imaging);New relapses, which are the occurrence of new symptoms or worsening of old symptoms.

In this study, we used data from a previous trial on the efficacy of the interferon beta-1a treatment of MS compared to a placebo. This trial is registered on ClinicalTrials.gov as NCT00906399. Information regarding standard protocol approvals, registrations, and patient consents is described by Calabresi et al. [25].

### 2.2. Inspiration for a New Method of Human–Algorithm Collaboration

In the context of a research project on clinical decision support systems (CDSSs), we have developed a functional interactive prototype of a CDSS for multiple sclerosis [26]. The CDSS was based on knowledge acquired through the observation and analysis of medical practices. It allows one to visualize how a given patient called POI (patient of interest) is positioned in the context of its patients of reference (PORs). PORs are a dataset of patients representing similar characteristics to the POI according to features identified in the literature and by MS experts. Through the projection of PORs data and direct interaction with the dataset using statistics, the CDSS displays potential outcome predictions for the predictive factors of disease activity (see Figure 1). It compares the prediction of the disease course according to different treatment options. We performed an evaluation of the developed CDSS with four MS experts. Participants were asked to use the CDSS to measure its usefulness in supporting medical decision-making. The results of the survey did not reveal any comments or concerns about the predictive method. Instead, participants were enthusiastic about the opportunity to interact with the CDSS and the fact that the PORs selection method was based on physicians’ reasoning, which they perceived appropriate to contextualize the POI.

Despite the success of predictive methods using statistics, other methods using machine learning still raise concerns from physicians.

This experience led us to think about a new method based on physician–algorithm collaboration to address physicians’ concerns about the potential consequences of widespread use of machine learning algorithms in clinical practice.

### 2.3. New Method for Hybrid Physician–Algorithm Intelligence

We proposed a new method for the prediction of disease progression through the development of hybrid predictive models leveraging computational power, good quality clinical data, and the interaction with expert physicians [27]. We were especially interested in the prediction of disease evolution made by physicians thanks to measurements (such as an MRI), medical data (such as an EDSS calculation), and direct observations of the patient (consultations). The proposed method aimed at integrating the physician in the prediction process, more precisely in the phase of the preparation of the data for learning as illustrated in Figure 2. Rather than using the values of a patient feature and categorizing it automatically, we proposed to instead use the equivalent group according to the physicians’ reasoning. For example, for the feature ‘age at disease onset’, the patient was 30 years old. According to the physicians’ reasoning, this patient belonged to group A2. Hence, instead of using the raw value of this feature, we would use the equivalent group. We would apply the same data transformation principle for the remaining features to all patients in the database. Then, the transformed data would be used to create a predictive model of the indicators known to predict disease activity according to the physicians’ reasoning. The categories of patient features used are described by C. Ed-driouch et al. [26].

### 2.4. Evaluation Method of the Hybrid Human–Algorithm Intelligence Method

To evaluate the proposed approach, we used data from a clinical trial (NCT00906399) to create predictive models of the following indicators of MS activity:EDSS: disability progression;New lesions: the appearance of new lesions on the MRI.

For each of these factors, we applied the prediction model creation and selection pipeline (Figure 3) described below to develop three types of prediction models based on different data preprocessing:All features: a model created using all existing features in the database—the usual brute force method;Physician features: a model created based on features used by MS physicians to make a medical decision—the first step in acknowledging human experts’ skills;Physician features & classes: a model created based on the categories of features used by MS physicians to make a medical decision—a more advanced analysis of human experts reasoning as they are categorizing patients when trying to predict MS course. Instead of using the raw patient data to train the model (a patient age at onset of 29 years old), the equivalent category from the physicians’ reasoning is used (a patient in the [20–40] category).

### 2.5. Pipeline for Creating and Selecting Prediction Models

#### 2.5.1. Data Separation

The first step of the pipeline was to separate the features that would be used by the model to make the prediction from the targets to be predicted (see step 1 in Figure 3). We used data at baseline to create prediction models of two indicators of MS activity one year later. The target indicators were the worsening of the EDSS and T2 lesions described in Table 1.

#### 2.5.2. Data Preprocessing

ML estimators can give inaccurate results if the individual features do not more or less match standard, normally distributed data. We noticed that the features in our dataset span different ranges (see Appendix A). Thus, we used transformers to normalize our data (see step 2 in Figure 3). We used StandardScalar, which shifts and scales each feature individually so that they all have a mean of 0 and a unit standard deviation.

Two different transformers were used to code the categorical data. The choice of transformer depends on the underlying model and the type of categories (nominal/ordinal). In general, OneHotEncoder is the encoding strategy used when the models are linear. It prevents models from making a false assumption about the order of categories by creating as many new columns as there are possible categories. Whereas OrdinalEncoder, which produces ordinal categories (e.g., 0 < 1 < 2), is used with tree-based models.

#### 2.5.3. Classifiers and Hyperparameter Tuning

Class imbalance is a typical problem in multiple sclerosis research because the disease progresses slowly. In consequence, datasets often contain many more records related to the ‘non-worsening’ class than to the ‘worsening’ class (see Table 2), which can lead to biased models if the problem is not handled properly. To reduce this bias, we used the following methods (see step 3 in Figure 3):

Different prediction models:

The performances of the prediction models differed according to the distribution of the data and the classes to be predicted (see Appendix A). We trained different prediction algorithms to choose the best performing prediction model from the created models. Three classical algorithms were chosen as a baseline [28]:Support vector classifier (SVC);Logistic regression;Decision tree;

and seven models of the ensemble learning technique [29] which consists of training several entities (estimators) of the same algorithm:In parallel, on a random portion of the dataset, to obtain diversified models since they were not all trained on the same data: bagging and RandomForest algorithms;In series, by asking each model to try to correct the errors made by its predecessor: Adaboost, GradientBoosting, and Xgboost algorithms;In combination, by training different algorithms to combine their results as new features to train a meta-classifier (stacking algorithm) or to predict the final result based on their combined majority of votes (voting algorithm).

We used the Adaboost and bagging algorithms 3 times, each time using one of the baseline models as the baseline estimator.

2.Weighting;

For the baseline, bagging, RandomForest, Adaboost, and XGBoost models, we controlled the balance of positive and negative weights in order to penalize errors on the minority class by a weight proportional to its underrepresentation.

3.Hyperparameter tuning and nested cross-validation.

Performing hyperparameter optimization can produce the best results compared to using the default settings to classify unbalanced datasets [30]. We applied 30 iterations of the ‘RandomizedSearchCV’ approach [31] to find the best parameters for each algorithm (except for stacking and voting algorithms, as we used them to perform meta-validation) based on our dataset.

Nested cross-validation provided a way to reduce bias in the combined setting of hyperparameters and model selection. Specifically, we performed external 10 folds cross-validation that divides the data into ten stratified folds to create folds with approximately the same proportions of examples from each class as the full data set. Each of the ten folds was then considered test data while the remaining folds were the training data. For each training set, we applied a stratified 5 folds cross-validation that was repeated 4 times to select the hyperparameters with the ‘RandomizedSearchCV’ method based on the highest ‘balanced accuracy’ score: the average of the proportion of correct responses from each class individually. As a result, for each model, we obtained 10 different sets of parameters. We restricted ourselves to the parameters of the model associated with the maximum value of the score ‘balanced accuracy’.

#### 2.5.4. Performance Evaluation

The models were re-evaluated (see step 4 in Figure 3) with a stratified 5 folds cross-validation, were filtered according to the mean of the AUC score (evaluation metric to measure the ability of a classifier to distinguish classes), and used ± a given standard deviation value.

#### 2.5.5. Meta-Evaluation

A meta-evaluation of the models was applied (see step 5 in Figure 3). The models obtained by the previous filtering were used as estimators for the stacking model. A meta-filtering similar to the previous one has been applied to decide whether to keep the stacking among the best performing models. The voting algorithm was also applied with the models obtained by the meta-filtering as estimators.

#### 2.5.6. Best Classifier

The best prediction model was the one with the maximum mean value of the area under the ROC (receiver operating characteristic) curve (AUC) [32] and the minimum value of its standard deviation.

## 3. Results

To evaluate the proposed method, we used data from patients on placebo (no treatment) and peginterferon beta-1a with FDA (Food and Drug Administration) approved doses. Data from 779 patients at baseline were used to create a predictive model for indicators of MS activity one year later. For each indicator (target), we created three models:All features: model created based on all existing features in the dataset;Physician features: model created based on the features used by MS physicians to make a medical decision;Physician features & classes: model created based on the categories of features used by MS physicians to make a medical decision.

We applied the prediction model pipeline described above (Figure 3) to create and select the three types of models for each target. We compared the performance of the three types of models for each target (see more details in the Appendix A). We used AUC and F1 scores to compare the final results as described in the following subsections:F1: measures the ability of a model to predict positive individuals well, both in terms of accuracy (rate of correct positive predictions) and recall (rate of correctly predicted positives);AUC: measures the ability of a classifier to distinguish classes.

### 3.1. Prediction Models of EDSS Worsening

Figure 4 shows the performance of the selected models for each type of model created for the prediction of EDSS worsening. For the ‘All features’ model, we found a mean AUC score of 0.53 and a mean F1 score of 0.66. In comparison, the performance of the model selected for the ‘Physician features’ method increased by +0.03 in the AUC score and +0.05 in the F1 score. A more important difference was found when comparing the performances of the ‘All features’ model to the ‘Physician features & classes’ model; the AUC score increased by +0.07 and the F1 score by +0.08 with lower standard deviation values (std values in Figure 4).

### 3.2. Prediction Models of T2 Lesions Worsening

Figure 5 shows the performance of the selected models for each type of model created for the prediction of lesions worsening. The AUC and F1 scores of the model based on the ‘All features’ method were 0.72 ± 0.04. Similar results were obtained for the models based on the ‘Physician features’ and ‘Physician features & classes’ methods with differences of only 0.01. The standard deviation of the AUC and F1 scores of the ‘All features’ model was 0.01 greater than the standard deviation of the AUC and F1 scores obtained for the ‘Physician features’ model: AUC and F1 scores = 0.73 ± 0.03. The means of the AUC and F1 scores of the ‘All features’ model were 0.01 less than those of both ‘Physician features’ model and the ‘Physician features & classes’ model: AUC and F1 scores = 0.72 ± 0.03.

## 4. Discussion

This paper presents an innovative method for developing hybrid predictive models [27] that leverages the power of ML algorithms, good quality data, and interactions with expert physicians to address the challenges of the adoption of ML in medical practice. Our use case concerns the prediction of Multiple Sclerosis (MS) course. The hybrid and collaborative nature of this work is two-fold; it combines the informed selection of clinically relevant variables and the interaction with ML algorithms at the bedside.

Applications of ML to predict the MS course have been discussed in recent reviews by S. Denissen et al. [33] and L. Hone et al. [34]. Only A. Tacchella et al. [35] have conducted a study about the prediction of the MS course based on the collaboration of human intelligence and ML algorithms. The results of the study have shown an important improvement of the prediction capacity obtained when the predictions were combined with a weight that depends on the consistency of the human (or algorithm) predictions. The hybridization of human and artificial intelligence in this study occurs at the prediction phase; hence, the prediction model is created independently of human intelligence. However, a good understanding of the data can lead to very important discoveries. Therefore, it is essential to consider the knowledge of the human domain expert as a main component of the overall process [36].

Our hybrid prediction method is based on the integration of physician reasoning in the creation process of the prediction model. We proposed the use of good quality clinical data that matches the purpose of the prediction model. For example, to predict the evolution of MS according to the treatment efficacy, we used data from a clinical trial concerned with this treatment and its effectiveness compared to a placebo for MS patients. In such a case, we are sure to train a model with data that reflects the clinical application. To facilitate understanding of the model learning process and to guide the model decision based on medical practice, we relied on the reasoning of expert physicians to select and categorize the important features. We conducted literature searches and interviews with MS physicians to select features and define their categories [26]. Instead of using raw patient data to train the model, we used categories for each feature. For example, for the cumulative lesion feature, we used two categories: ‘patients with less than 10 lesions on MRI’ and ‘patients with 10 or more lesions on MRI’. This reflects the physicians’ expertise that the disease course in patients with 10 or more lesions on MRI is different from that of patients with less than 10 lesions. The categorized data is then used to train the ML algorithm. This ensures that the reasoning of the created prediction model is included in the decision-making process of the medical practice, as it is based on the knowledge and experience of expert physicians.

To evaluate the method, we developed a detailed prediction model pipeline that addresses the class imbalance problem, which is a typical problem in MS. We applied the pipeline to create prediction models of the EDSS and lesions worsening. For each target, we selected three types of prediction models: (1) the ‘All features’ model, which is based on all features present in the dataset; (2) the ‘Physician features’ model, which is based on only the features used by the expert physician to make a decision; and (3) the ‘Physician features & classes’ model, which is based on the features and the physician’s categorization. We compared the performance of the models obtained for each of the targets.

Despite the large imbalance between EDSS classes (no worsening for 728 patients and worsening for 51 patients), an important difference was found between the different types of models, especially between the ‘Physician features & classes’ model and the ‘All features’ model. The mean AUC score of the ‘Physician features & classes’ model is 4% higher than the ‘Physician features’ model and 7% higher than the ‘All features’ model. The mean F1 score of the ‘Physician features & classes’ model is 3% higher than the ‘Physician features’ model and 8% higher than the ‘All features’ model. The difference between the three types of the prediction models shows that the performance of the ‘Physician features & classes’ prediction model is the best, which validates the usefulness of our hybrid prediction method. For the prediction of lesions worsening, we found that the three types of models had approximately the same performance. This confirms that it is sufficient to rely on the reasoning of the expert physician to create prediction models. In addition, models that include the physician’s reasoning are more reassuring because they guide the decision of the prediction model based on medical practice (the experience of the expert physician).

If available in the dataset, other features such as cognition, quality of life, predictive biomarkers, etc. could be used to train the ML algorithms.

In future work, we will develop an interactive real-time CDSS that allows the physician to directly modify the algorithm’s internal decision process through synchronous interaction with the system. In addition, the system has to be clear in terms of the quantity and quality of the data and the way the proposed results are obtained and displayed.

## 5. Conclusions

Several ML solutions have been developed to address the complexity of chronic diseases. However, the adoption of these solutions in healthcare systems is still limited. The future of the ML algorithm certification and the actual use of these emerging tools in daily clinical practice remain challenging. To promote their use, ML solutions need the support of expert humans. Thus, they will be driven by human experience and, together, will bring out their greatest potential. In addition, this can facilitate understanding and acceptability, as expert human reasoning is included in the process of ML.

We have proposed a new hybrid physician–algorithm intelligence method that integrates empowered humans into the learning process of ML algorithms. We promote the use of quality-approved data to be transformed according to the knowledge of expert physicians and then used to train ML algorithms. The results of the comparison between the conventional method of creating prediction models and our HI method showed that the models based on the latter can perform better.

The hybridization of human and artificial intelligence offers the most promise to overcome the barriers that limit the implementation of ML-based systems in healthcare and foster a safe and revolutionary future of medicine. The proposed method is intended to enable reliable and acceptable applications of ML algorithms to support the personalization of clinical decision-making.

## Figures and Tables

**Figure 1 sensors-22-08313-f001:**
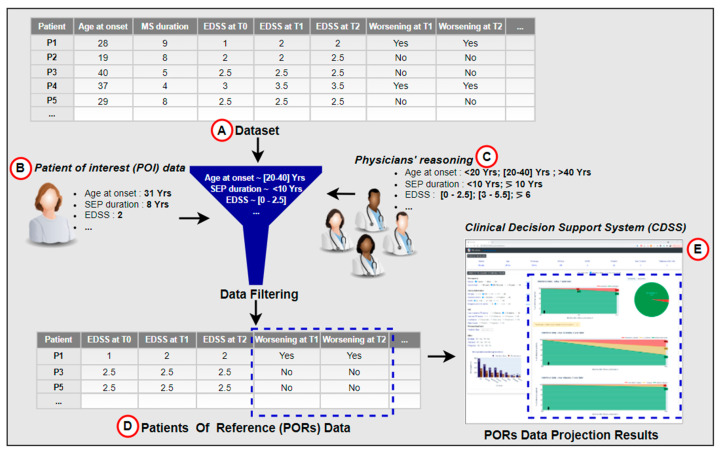
Data projection as inspiration for a hybrid prediction approach. The data projection process relies on filtering the dataset (**A**), according to the POI’s data (**B**), provided by the physician, and embedded expert physicians’ knowledge (**C**). The filtered data (PORs) in (**D**) are further used to visualize the POI’s potential disease course, seen in (**E**).

**Figure 2 sensors-22-08313-f002:**
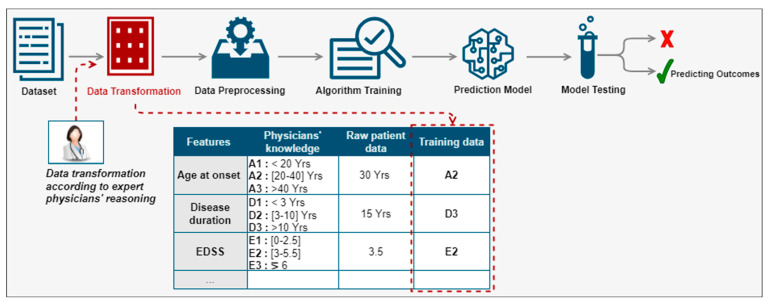
Physician reasoning integration in the prediction process through the data transformation red block, before training the prediction model (see details in Figure 3): rather than using the values of a patient feature, we use the equivalent group according to the physicians’ reasoning.

**Figure 3 sensors-22-08313-f003:**
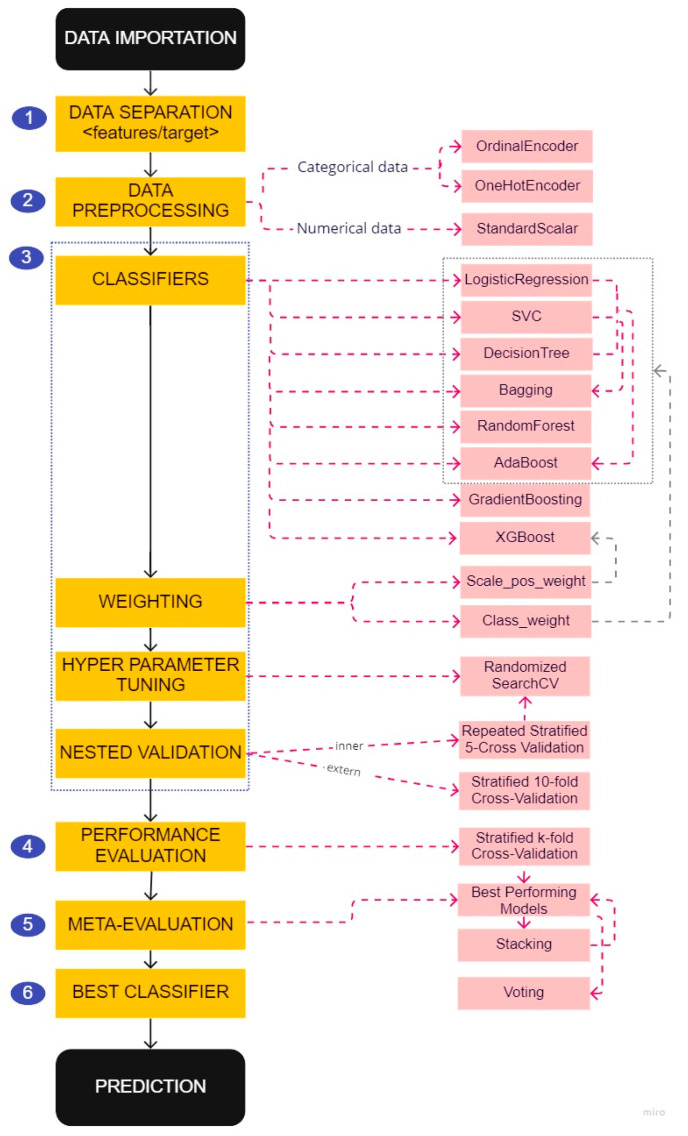
Pipeline for the creation and selection of predictive models.

**Figure 4 sensors-22-08313-f004:**
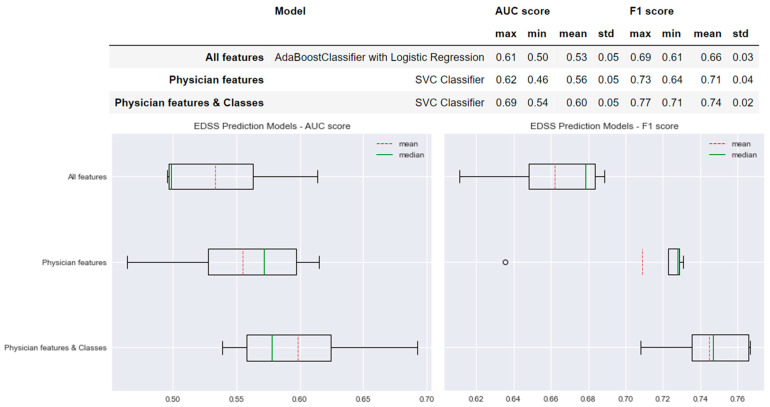
Selected predictive models and the results obtained for AUC and F1 scores for each type of model created for EDSS prediction.

**Figure 5 sensors-22-08313-f005:**
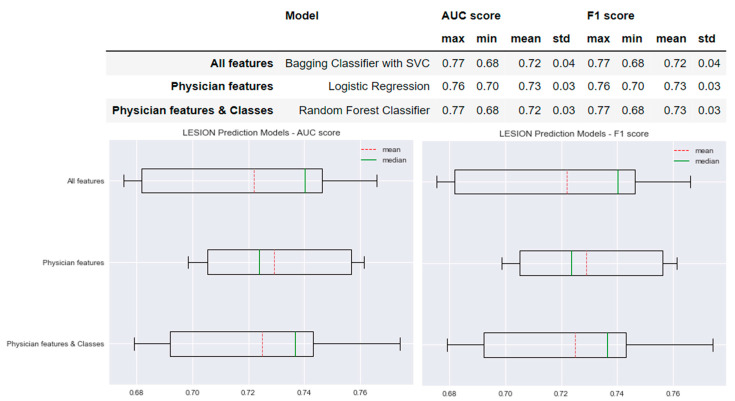
Selected predictive models and the results obtained for AUC and F1 scores for each type of model created for New Lesion prediction.

**Table 1 sensors-22-08313-t001:** Explanation of the worsening of the target factors.

Target	Worsening
EDSS	Progression of 1 point for EDSS between [1–5.5]
	Progression of 0.5 point for EDSS ≥ 6
	Progression of 1.5 point for EDSS = 0
T2 Lesion	Appearance of new lesions on the MRI

**Table 2 sensors-22-08313-t002:** Proportion of each class for the factors to be predicted.

Target\Class	Non Worsening	Worsening
EDSS	728	51
T2 Lesion	394	385

## Data Availability

Not applicable.

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
