# Peer review of "Addressing the Challenges and Barriers to the Integration of Machine Learning into Clinical Practice: An Innovative Method to Hybrid Human–Machine Intelligence"

_sensors, 2022, doi:10.3390/s22218313_

Round 1

Reviewer 1 Report

A hybrid human-machine intelligence method is proposed in this paper to promote the use of ML algorithms in healthcare systems. This is a valuable research topic. However, some key issues need to be addressed before publication.

1.      The structure of this paper is inappropriate.  

a)        Research on the application of machine learning in healthcare, especially the research related to hybrid human-machine intelligence, should be introduced in Introduction Section, to highlight the contribution of this paper.

b)        However, Research related to hybrid human-machine intelligence is introduced in Section Discussion, which does not make sense.

c)        It would be better to have a Conclusions Section.

2.      In Section 2.1, the proposed method aims at integrating the physician in the prediction process. This is realized by using the equivalent group of patient feature according to the physicians' reasoning, which is called “data transformation principle” in Line 158. This is the main contribution of this paper. However, in Section 2.5.2, the transformation in step 2 in Figure 3 just normalize data. There is no " physicians 's reasoning" in Figure 3. Please explain it.

3.      In Section 2.5.3, to solve the problem of biased models caused by imbalanced data, the authors used the following methods: “Different prediction model”, ”Weighting”, and ”Hyperparameter tuning and Nested cross-validation”.

a)        However, the above three methods are the process of training the model. It seems that only the ”Weighting” method is useful to reduce the bias of models. Please explain it.

b)        Besides, the above three methods are not consistent with step 3 in Figure 3. The “Weighting” is missing in Figure 3.

4.      There are some other minor issues.

a)        Line 226, what the meaning of SVC? Do you mean Support Vector Classifier? Please spell the first abbreviation.

b)        Line 229, do you mean “seven” or “several” ?

c)        A variety of machine learning algorithms are adopted in this paper. The algorithms used for different data are different. Maybe a table to list them is better.

Reviewer 2 Report

The work titled ‘Addressing the challenges and barriers to the integration of machine learning into clinical practice: An innovative method to hybrid human-machine intelligence’, a hybrid human-machine 20 intelligence method to promote the use of ML algorithms in healthcare systems is presented. This work need to address some address revisions/concerns before final publication.

1. What is novelty of the work. Please underscore the scientific value added/contributions of your paper in your abstract and introduction and address your debate shortly in the abstract.

2. A good article should include, (1)originality, new perspectives or insights; (2) international interest; and (3) relevance for governance, policy or practical perspective.

3. The work is devoted to an actual scientific and applied problem, performed by correct modern methods and the results are not in doubt. But the presentation and discussion of the results, as well as the conclusions, need to be improved.

4. Figure 1. The data projection process relies on the use of physicians’ reasoning and patient's features  to filter the dataset and obtain the PORs of a given patient. The PORs data is further used to visualize the possible evolution of the predictive factors of the disease activity. What is the significance of this figure?

5. Literature review section need to be enhanced. Add some recent related works. Some suggested works are :

Wang, F. Y., Guo, J., Bu, G., & Zhang, J. J. (2022). Mutually trustworthy human-machine knowledge automation and hybrid augmented intelligence: mechanisms and applications of cognition, management, and control for complex systems. Frontiers of Information Technology & Electronic Engineering23(8), 1142-1157.

Akmeikina, E., Eilers, K., Li, M. M., & Peters, C. (2022, July). Empowerment Effects in Human-machine Collaboration-A Systematic Literature Review and Directions on Hybrid Intelligence Behavior Patterns. In Pacific Asia Conference on Information Systems (PACIS).

Fu, S., Qin, D., Amariucai, G., Qiao, D., Guan, Y., & Smiley, A. (2022, May). Artificial Intelligence Meets Kinesthetic Intelligence: Mouse-based User Authentication based on Hybrid Human-Machine Learning. In Proceedings of the 2022 ACM on Asia Conference on Computer and Communications Security (pp. 1034-1048).

Tiwari, S., Jain, A., Sapra, V., Koundal, D., Alenezi, F., Polat, K., ... & Nour, M. (2022). A Smart Decision Support System to Diagnose Arrhythymia using Ensembled ConvNet and ConvNet-LSTM Model. Expert Systems with Applications, 118933.

5. Figure 3. Pipeline for the creation and selection of predictive models. Is this pipeline generic.

6. Compare the results with other state-of-the-art methods.
